# From struggle to strength in African and Middle Eastern newcomers' integration stories to Canada: A participatory health equity research study

Maggie Fong[1], Amy Liu[2], Bryan Lung[3,4], Ibrahim Alayche[5], Shahab Sayfi[6], Ryan Yuhi Kirenga[7], Marie Hélène Chomienne[8,9,10], Ammar Saad[9], Jean Grenier[8,10,11], Azaad Kassam[12], Rukhsana Ahmed[13], Kevin Pottie[10,14]*

1 Department of Epidemiology and Biostatistics, Western University, London, Ontario, Canada, 2 Department of Interdisciplinary Medical Sciences, Western University, London, Ontario, Canada, 3 Department of Anatomy and Cell Biology, Western University, London, Ontario, Canada, 4 Department of Biochemistry, Western University, London, Ontario, Canada, 5 Department of Medicine, University of Ottawa, Ottawa, Ontario, Canada, 6 Department of Health Research Methods, Evidence and Impact, McMaster University, Hamilton, Ontario, Canada, 7 The Vulnerability, Trauma, Resilience and Culture Research Laboratory (V-TRaC), University of Ottawa, Ottawa, Ontario, Canada, 8 Department of Family Medicine, University of Ottawa, Ottawa, Ontario, Canada, 9 School of Epidemiology and Public Health, University of Ottawa, Ottawa, Ontario, Canada, 10 Institut du Savoir Montfort, Ottawa, Ontario, Canada, 11 School of Psychology, University of Ottawa, Ottawa, Ontario, Canada, 12 Department of Psychiatry, University of Ottawa, Ottawa, Ontario, Canada, 13 Department of Communication, University at Albany, State University of New York, Albany, New York, United States of America, 14 Department of Family Medicine, Western University, London, Ontario, Canada

* kpottie@uwo.ca

**Data Availability Statement:** Data cannot be shared publicly because of ethical concerns. Minimal data set underlying the results described

## Abstract

### Background and objectives

Newcomers (immigrants, refugees, and international students) face many personal, gender, cultural, environmental and health system barriers when integrating into a new society. These struggles can affect their health and social care, reducing access to mental health care. This study explores the lived experiences of African and Middle Eastern newcomers to Ontario, Canada. An understanding of newcomer integration challenges, successes and social justice issues is needed to improve health equity and social services.

### Methods

In this qualitative study, we used a participatory research approach to collect stories reflecting participants' integration perspectives and experiences. Beginning with our immigrant community network, we used snowball sampling to recruit newcomers, ages 18 to 30, originating from Africa or the Middle East. We used qualitative narrative analysis to interpret stories, identifying context themes, integrating related barriers and facilitators, and resolutions and learnings. We shared our findings and sought final feedback from our participants.

in our manuscript has been provided in the supporting information files. Complete data are available from the Montfort Hospital Institutional Data Access / Ethics Committee (contact Marie-Andrée Imbeault via email at marieaimbeault@montfort.on.ca) for researchers who meet the criteria for access to confidential data.

**Funding:** The study was financially supported by the Institut du Savior Montfort (1920-HQ-000305) at the Montfort Hospital, Ottawa, Onartio, Canada. The funders had no role in the study design, data collection and analysis, decision to publish, or preparation of the manuscript.

**Competing interests:** The authors have declared that no competing interests exist.

## Findings

A total of 18 newcomers, 78% female and approximately half post-secondary students, participated in the study. Participants described an unknown and intimidating migration context, with periods of loneliness and isolation aggravated by cold winter conditions and unfamiliar language and culture. Amidst the struggles, the support of friends and family, along with engaging in schoolwork, exploring new learning opportunities, and participating in community services, all facilitated integration and forged new resilience.

## Conclusions

Community building, friendships, and local services emerged as key elements for future immigrant service research. Utilizing a participatory health research approach allowed us to respond to the call for social justice-oriented research that helps to generate scientific knowledge for promoting culturally adaptive health care and access for marginalized populations.

## Introduction

In 2021, Canada's population had its largest proportion of immigrants in over 150 years, with the proportion of those from Africa and the Middle East steadily rising [1]. Ontario received over 40% of all immigrants to Canada [2]. Currently, in Canada, more than a tenth of newcomers (immigrants, refugees, and international students) are 18–30 years of age and it is expected that 30–49% of the total Canadian population will consist of children and youth with an immigrant background by 2036 [3]. Therefore, it is critical that Canada's health care system be accessible and provide culturally adaptive care to help young newcomers improve both their physical and psychological well-being. Research suggests there are unmet needs to access health care specific to immigrants in Canada [4]. The likelihood of accessing mental health care among newcomers also varies depending on their world region of origin [5]. Despite a growing population of newcomer youth in Ontario, comprehensive understandings of newcomer youth experiences and their integration to Ontario are underdeveloped, especially for Ontarian newcomer youth from Africa and the Middle East [6].

Young newcomers can be particularly vulnerable to mental health issues, as they often face additional personal, gender, cultural, environmental, and socioeconomic barriers when integrating into a new society [7]. Resettlement challenges can impact the mental health of newcomers and further reduce access to appropriate resources and care [8]. There is a need to understand factors that challenge and guide young newcomers' integration journeys in Ontario and their implication for community and mental health care resources. An understanding of the lived experiences of newcomers through their integration challenges is, therefore, needed to improve access to and uptake of mental health services [9,10].

We used a community-based participatory research approach to reduce racial and ethnic health inequities and to influence health service policy [11]. This project is part of a larger participatory health research (PHR) program with refugees and other migrants at the Canadian Collaboration for Immigrant and Refugee Health (www.ccirhken.ca). Trust is recognized as a key component to story collection [12,13], and thus we used our existing youth leaders from the CCIRH to begin the process and contribute to the research interpretation. Building on the success of narrative research approaches to study health professional integration [14], our objective was to capture stories that inform our understanding of our participants' early integration lived experiences.

## Methods

### Ethics statement

Written informed consent was obtained by all participants and the study was approved by the Institutional Review Board at the Montfort Hospital: File ID Number: 22-23-08-022.

### Methodology

We used a community-based participatory health approach [15] together with narrative analysis [12,16] to capture the lived experiences of newly arriving youth newcomers to Ontario, Canada and how their experiences magnified or reduced social injustice during their resettlement. We purposefully sampled newcomers from Africa and the Middle East to understand participants coming from some socio-politically unstable regions. Our culturally diverse research team included several newcomer youths with immigration-related experiences, along with communication, cultural psychiatry and primary care expertise. Our research team emerged from existing Canadian Collaboration for Immigrant and Refugee Health co-design sharing circles and the intent to both work with and potentially help newcomer youth as part of the evolving PHR approach. We reflected as we worked together on our story project and decided to embark on a formal story research project. Three young community leaders (AS, SS, RYK) played a key role in recruiting participants and building trust throughout the study period and thereafter. We performed an inductive qualitative analysis of newcomer stories to investigate the challenges, lived experiences, and successes newcomers face when they first arrive in Ontario.

### Participant recruitment and eligibility

Participants were informed about our study through an infographic invitation poster shared by our colleagues from the University of Ottawa and Western University. Beginning with our community youth leaders/co-researchers, snowball sampling was used to recruit participants. To be eligible for this study, a newcomer had to be: between 18 to 30 years of age; either an immigrant, refugee, or international student; originated from Africa or the Middle East and arrived in Ontario within the last 10 years; were affiliated with an Ontario institution or non-governmental organization; able to read and write in English or French; and able to provide informed consent. For psychological safety reasons for our participants, we did not include migrants within their first year of arrival, and we selected migrants with some existing community ties to reduce risks and ensure existing support. Our newcomer youth leaders played a key role in building trustworthy relationships with potential participants, and this trust was essential to inspire and collect authentic stories from vulnerable youth.

### Research activities

For this qualitative study, our core framework consisted of a community-based participatory approach [15]. We supported our three young community leaders in recruiting participants into the project and building trust with them throughout. Co-researchers with lived experience immigrating from Africa or the Middle East were involved in each stage of the research project: design, recruitment and data collection, analysis, and writing. Study data were collected and managed using REDCap electronic data capture tools hosted at the Montfort Hospital. Participants were required to complete a questionnaire collecting information about their age, gender, country of origin, migration status, education level, French/English proficiency, and student versus employment status. Participants were provided with an e-learning module on storytelling, which outlined prompts and story examples to guide participants. Participants

then shared a written story about their integration experience and journey. The written stories were anonymized by removing all identifiable information. All French stories were translated by a bilingual co-researcher (BL) and validated by another (IA) to mitigate language translation biases. We sought a range of feedback from participants as part of the research process.

## Analytical approach

Four youth researchers (MF, AL, BL, IA) were trained and supervised in narrative qualitative analysis by two researchers (RA, KP). The primary analysts were undergraduate students who were similar in age to some participants. Prior to data collection, researchers wrote stories based on experiences as second-generation immigrants and from witnessing those around them. This exercise allowed researchers to reflect on their perspectives and potential reflexivity. The researchers independently immersed, read, and reflected on the collected narratives. They used inductive analysis to identify themes within the framework of a narrative synthesis [16]: context, barriers, facilitators, and resolution. The themes were agreed on within the larger research team, and all stories were coded accordingly by each youth researcher. The analysts then extracted representative quotations for the themes. Disagreements in interpretation were resolved through discussions with the larger research team. We collected stories until new themes were no longer arising (saturation). For credibility, the summary of themes and quotations was shared with our youth leader co-investigators and our general group of participants for member checking to validate our interpretation of participant experiences. Finally, we sought feedback from participants on the impact of their story writing experience.

## Results

### Study participants

Eighteen youth newcomers participated in the study, sharing eighteen written stories. The characteristics of the participants are presented in Table 1. In summary, fourteen participants (77.8%) identified as females, whereas only four (22.2%) identified as males; the majority were students (55.6%) and immigrated from Africa (72.2%). Most participants completed some form of high school or higher education. English was the primary language most participants preferred for both communication and medical care (88.9%). However, most of the French-speaking participants were also proficient in English.

### Context

**(i)** *Academic institution*: Some participants felt that integrating into a new school system can be very challenging at first. Some challenges included not knowing where a newcomer belongs and feeling isolated within the large classroom sizes of the Canadian postsecondary system. However, with time and support from school resources, school became a great opportunity to feel a sense of responsibility and foster knowledge and skills towards pursuing ambitions. School was also perceived to serve as a starting point for young newcomers towards establishing meaningful relationships with other young people in Canada.

> "*What struck me on my first day of university was how little interest people had in talking and making new relations. Where I come from, university classes were very similar to high school classes. . .Classes here had more than 200 students in them, and so people just wanted to attend the class and leave. That definitely led to a sense of isolation at first, as I also had trouble knowing what to say and to whom.*" *-Kemi, a male from Africa*

**Table 1. Participant characteristics.**

| Characteristics | | Frequency (n = 18) | Percentage (%) |
|---|---|---|---|
| **Gender** | Female | 14 | 77.8 |
| | Male | 4 | 22.2 |
| **Area of Origin** | African | 13 | 72.2 |
| | Middle-Eastern | 5 | 27.8 |
| **Immigration Status** | International Student | 9 | 50.0 |
| | Economic Immigration | 4 | 22.2 |
| | Refugee | 2 | 11.1 |
| | Other | 3 | 16.7 |
| **English Proficiency Level** | Basic | 1 | 5.6 |
| | Proficient | 15 | 83.3 |
| | Native Language | 2 | 11.1 |
| **French Proficiency Level** | Basic | 5 | 27.8 |
| | Intermediate | 3 | 16.7 |
| | Proficient | 7 | 38.9 |
| | Native Language | 2 | 11.1 |
| | Prefer not to Answer | 1 | 5.6 |
| **Preferred Language when Receiving or Engaging Mental Health Services** | English | 16 | 88.9 |
| | French | 1 | 5.6 |
| | Other | 1 | 5.6 |
| **Highest Level of Education Completed** | Grade 10 to 12 | 7 | 38.9 |
| | College Diploma | 3 | 16.7 |
| | Bachelor's Degree | 2 | 11.1 |
| | Master's Degree or Doctorate Degree | 4 | 22.2 |
| | Prefer not to Answer | 2 | 11.1 |
| **Employment Status** | Full-time Employment | 7 | 38.9 |
| | Unemployed | 1 | 5.6 |
| | Student | 10 | 55.6 |

*"The black community in my university was very welcomed and friendly. They showed me around campus and were there to answer any questions I had about the student experience in my university (how to get a bus pass, how to get a meal plan, etc.)." -Amna, a female from Africa*

**(ii)** *Work experiences*: Being a newcomer poses unique circumstances that make it difficult to have the same work opportunities as those born in Canada. For some participants, it was important to find a stable source of income upon arrival to Canada in order to support themselves and their families. However, despite knowing English and having work experiences in their country of origin, some experienced difficulties finding opportunities in Canada due to a limited social network. No stories directly suggested discrimination as a barrier. Multiple newcomers discussed cashiering as their first source of income. Nevertheless, once newcomers found meaningful work experience in Ontario, they appreciated the cultural immersion and meeting new people.

*"As an immigrant, it was hard for me to find employers who were willing to hire someone with no Canadian experience or references." -Sade, a male from Africa*

"*I decided to help out and get a job, but it was hard because I had no work experience in Canada*". -*Youssef, a male from the Middle East*

## Integration-related barriers

**(i)** *Winter weather conditions*: The frigid winters make adapting to Canada more difficult and isolating for newcomers. Oftentimes it can take a toll on newcomers physically, emotionally, and socially. The winter season in Canada felt very foreign to many participants, especially those from more tropical weather. With often gloomy weather, it was more difficult for many to take care of themselves. As the days were shorter and colder in the winter, the challenge to get out of home to exercise, get fresh air and meet new people was expressed in many of the stories. Therefore, achieving basic needs and feeling a sense of belonging during this time can seem more difficult for newcomers.

"*No one prepares you for the kind of cold you experience here in Canada especially so when you're moving from a warm country like Kenya. . . .The depression that winter brings is one that has been spoken about quite a lot in the form of seasonal depression. I too, was a victim to this shortly after landing. I was particularly enraged by the 'fake sun' which is what I'd call it when the sun was out, shining despite it being -30 outside.*" -*Fatima, a female from Africa*

"*My biggest challenge while living in Canada is the cold, which is unavoidable and can really take a toll on one's mental health.*" -*Kofi, a male from Africa*

**(ii)** *Culture and language barriers*: Cultural and language differences brought a sense of disconnect from new communities. Upon arrival in Canada, newcomers are often unfamiliar with English or French language expressions. Local idioms are common and are key to communicating effectively [17]. These disparities were perceived as challenges to communication and a source of confusion regarding expectations and norms. Thus, it was common for some participants to feel out of place or have difficulty making new connections.

"*One of the first incidents I noticed was language disparities; it seemed that the way I learned English was not practical enough. Adversely, I discovered many of my classmates are bilingual. My mentality was moving on a downward slope, and I was fueled by the belief that my journey had failed.*" -*Noura, a female from the Middle East*

"*When I first arrived in Canada, I felt overwhelmed by the language barrier. Although I had studied English in Nigeria, it was still hard for me to understand the Canadian accent and slang. This made it difficult for me to communicate with people around me and make new friends.*" -*Sade, a male from Africa*

**(iii)** *Isolation, loneliness, and loss of hope*: Lack of family, friends, and community fosters integration difficulties and a sense of loneliness for newcomers. Upon arrival to a new country, feelings of loneliness and isolation arose due to the distance from family and friends. Many participants expressed difficulties finding and creating new meaningful connections when they first arrived in Canada, resulting in mental health challenges.

"*The other thing that hit me so hard about this country is the loneliness that is so omnipresent. You can go a whole day without people ever acknowledging your existence.*"—*Thierry, a male from Africa*

"*My mentality was moving on a downward slope, and I was fueled by the belief that my journey had failed. After four months, stress, anxiety, and somehow depression overcame me. I*

*went back to my home country. I lost considerable weight in just four months, and this made my parents feel nervous about my mentality." -Noura, a female from Middle East*

**(iv)** *Lack of resources and unknown scary migration context*: For many participants, the transition to Canada was both thrilling and terrifying. Many found it scary not knowing what the future held for them in their new country. Oftentimes, this was intensified when newcomers were separated from their families and social circle. Many described not knowing where to obtain basic services when first settling down or not having the resources necessary to facilitate their integration. Some participants specifically expressed challenges with locating and accessing mental health services. Beyond the fears, many also found the transition exciting when thinking about the new opportunities that may instill them in this new country.

"*There is so much about Canada that no one never really warns you about. It's the unknown of it all that made my transition to Canada so daunting." -Thierry, a male from Africa*

"*Before coming to Canada, I was scared about how my experience would turn out. Moving to a whole new country can be scary and you actually never know what to expect." -Amina, a female from Africa*

## Integration-related facilitators

**(i)** *Social support*: Having family and friends to support newcomers' new journey into Canada helped ensure a smooth transition into a new society. Unfortunately, those that did not have this level of social capital found it challenging to resettle. However, one of the primary objectives of immigrating to Canada, for many, was to establish a social circle and create meaningful connections with individuals who share similar interests.

"*I'll always be thankful for having relatives in this country because they make my experience easier. When I have a problem, or miss my family, I can just call them and they'll always be there for me." -Amina, a female from Africa*

"*I had a couple of friends here who supported me and gave me all the guidance I needed." -Amara, a female from Africa*

**(ii)** *Diversity*: Participants discussed their appreciation for multiculturalism, which was first experienced by many in Canada. Diversity and pluralism (an ethic of respect for diversity, viewing diversity as a strength and putting it into action) found in Canada was a major facilitator to help newcomers integrate into Canadian society. The ability to express one's own cultural beliefs and practices in the absence of one's own community made Canada more safe and welcoming. This might be the result of newcomers being able to find individuals who share their culture, norms, values, and hobbies, allowing these newcomers to bond with people more easily.

"*At the same time, moving here can be so exhilarating, exciting even because you get to start a fresh, get to experience new things, different people and cultures" -Fatima, a female from Africa*

"*My positive moments were the cultural diversity I felt in Canada, work opportunities, and all the activities that I could do over here." -Ayana, a female from Africa*

**(iii)** *New opportunities*: Discovering new opportunities, whether that be for work or for higher education, helped participants grow and learn from others around them. Immigrating

to Canada was perceived to advance newcomers' career paths and facilitate providing better support for themselves or their families. Many were confident in their ability to find a balance between school or work and their personal life, which led them to explore hobbies and interests.

> "*Another positive moment was being able to pursue higher education opportunities in Canada that weren't available in Nigeria. With access to quality education, I was able to further my studies and gain valuable knowledge that has helped shape my career path today.*" -Sade, a male from Africa

> "*I have recently learned to find ways to interact with people here. I joined a boxing club and made many new friends there, friends from all over the world.*" -Thierry, a male from Africa

### Resolutions and learning

*(i) Hope and needing time to adapt*: Newcomers' perspectives around the accessibility and ability to ask for support, a sense of belonging, and the endless opportunities that once seemed impossible to reach, brought happiness and hope. Time, patience, and the willingness to accept change allowed many participants to embrace the new environment. In several stories, participants highlighted that their initial experiences in Canada were not always as they had hoped and that it typically required a period of adjustment. Many expressed the need to adapt and change their perspectives to improve their well-being and feel more comfortable in their new environment.

> "*In one year, Canada gave me a sense of belonging that I did not ever feel during my 28 years living in Lebanon. Adapting to the language, culture, and people is still something I work on every day. It is part of me growing as a person and opening up to this new side of the world.*"... "*I hope that in a few years, I will be able to confidently say "Canada is home".* -Cilia, a female from the Middle East.

> "*Things improved as I began to understand the dynamics and interactions of students here. I had to adapt and change to how I thought I should be.*" -Kemi, a female from Africa

*(iii) Developing new strategies and gaining resilience*: Becoming involved in community, whether through hobbies or programs offered on campus, helped participants connect with others and feel a sense of belonging (an internal sustained psychological state). Many participants emphasized the need to connect with others to develop a sense of belonging. To achieve this, participants mentioned that involvement at community centers helped them access resources and engage further with others. Indeed, many newcomers emphasized that they had to proactively develop strategies that would enable them to integrate smoothly with their community and social circle. Additionally, participants expressed that the resolution to persevere and the ability to learn from adversity enabled them to cultivate diligence and independence, transforming them into a better, more resilient version of themselves.

> "*I do believe that all those experiences, for better or worse, shaped me into the person I am today. I learned a lot about myself over the last few years, and so despite the challenges and the difficulties, I wouldn't change much. It taught me resilience and endurance, and that it was fine to be on your own for a while. It also taught me that I am much stronger than I believed I was, and those lessons I will carry with me wherever I go.*" -Kemi, a female from Africa

*"Overall, I'm very grateful for all the struggle I went through cause it taught me hard work and not depending on others." -Youssef, a male from Middle East*

## Discussion

This study has engaged and explored the lived experience of young people immigrating to Ontario from Africa and the Middle East. The findings of this study shine a light on key challenges and adaptive processes that play out in the lives of newcomers. We identified common struggles with isolation and loneliness related to Canadian winter, large and impersonal classrooms, and cultural, language and employment barriers. We identified transitional experiences that included finding and developing social connections and family support. Finally, the stories highlight the discovery of opportunities that may be associated with social health and engagement within a pluralistic society. Our participatory approach to story sharing allowed participants to reflect on their own challenges and adaptive means to overcome barriers in collaboration with sharing story themes with other participants. Sharing written stories required literacy, trust, resilience, and self-confidence, which are also key attributes that enable emerging leaders.

The most common barriers leading to a feeling of isolation in this study were winter weather conditions, cultural and language barriers, and a feeling of having limited integration resources. Feeling unprepared for weather and immigration transition challenges was a common lived experience for newcomers integrating into the Canadian context. The severe Canadian winter led to isolation and was identified as an imposing barrier to developing social connections, a challenge that is not always spoken about in settlement information. This can lead to feelings of loneliness and prevent newcomers from establishing a sense of belonging and home in their new environment. Winter weather as a prominent Canadian integration barrier is often not discussed in the literature. It would be important to place consideration on this barrier in the development of culturally competent services for newcomer youth. Ensuring that newcomer youth are equipped with the knowledge and resources to help them adjust to the Canadian winters prior to arrival may also mitigate fears and stresses that were commonly associated with the unknowns of moving to Canada.

Similar to our findings, various Canadian studies discuss getting accustomed to new languages, social isolation, and cultural identity confusion as common post-migration barriers for newcomer youth [7]. Young newcomers often discussed challenges in establishing social networks and meaningful connections in school [6]. Newcomer students often felt like outsiders due to the large size and lecture-style classes.

Newcomers who were able to develop supportive social networks in Canada often reported more positive academic experiences compared to those who reported feeling isolated. Our findings highlight the need for community social resources in the academic setting to support newcomers not only academically, but also socially and psychologically to ensure better productivity and well-being [9,18]. School and work environments made an early impact on newcomers' integration journeys. Student resources and clubs catering to newcomers and international students facilitated transition into their academic setting. Cultural diversity and pluralism in the workplace and school environment can also help improve integration experiences of newcomers and make it seem less intimidating. Therefore, many newcomers described their experiences as students, employees, or both as a prominent part of their identity and integration story in Canada. Programs that provide opportunities for youth to form peer relationships or connect with other newcomers to exchange information on support and integration would be beneficial [19].

Newcomer participants often referred to seeking new opportunities as positive situations that facilitated their integration into Canadian society. Newcomers who focused on opportunities appeared better able to address their anxiety in a new country. This interpretation may be justified by a qualitative study on social support for immigrants and refugees in Canada, in which newcomers valued social health and support since it fosters a sense of empowerment, community, social integration, building networks, sharing experiences/problems, reducing stress, and physical/mental distress [20]. Furthermore, another common theme among newcomer-written stories is the seeking of new opportunities in Canada. Based on a study on immigrant and refugee youth in Montreal, Canada, these individuals have encountered opportunities and successes that helped them adapt to living conditions when first settling down in Canada [8]. Accordingly, The Resilience Framework [21] also suggests that finding new opportunities also helps newcomers plan out their future, allowing them to achieve their final goal. Therefore, discovering new opportunities proves to be a strong facilitator that helps Canadian newcomers integrate into a new society.

Some stories discussed the challenges of focusing on other aspects of life, such as careers, ambitions, or education when basic living necessities are a concern. These findings support ideas of The Resilience Framework, which emphasizes the importance of prioritizing basic needs when trying to build resiliency within vulnerable populations [21]. Our findings highlighted the challenges many newcomers face with finding job opportunities when first arriving in Canada. Multiple newcomers discussed the frustration of not having their past work and academic experiences validated in Canada. This finding possibly suggests that limited employment among young newcomers were confounded by language barriers and limited validation of academic credentials [6,9].

Our findings strongly support previous recommendations for embedding more mental health support and other social services at the community level rather than clinically for newcomers [22]. This strategy helps to increase accessibility as well as reduces potential cultural stigma and mistrust in health professionals. More importantly, these resources and services should be newcomer-focused and newcomer-led. Ensuring newcomers within the communities are integrated into the decision-making and leadership processes is pivotal to both service users and the community as a whole [22]. Being able to meet other newcomers who have shared similar experiences helped facilitate participants to find a sense of belonging in their new work, school, and home communities. Recommendations by the Mental Health Commission of Canada have also emphasized, "a central part of each provincial and regional plan to improve the mental health of [immigrant, refugee, ethno-cultural, and racialized, or IRER] groups must include the involvement of IRER communities, consumers, and families in planning, decision-making, implementation, and evaluation" [22]. Similarly, organizations, including Canadian post-secondary institutions, that aim to better support young newcomers to Canada should ensure that youth have a voice and leadership role in the organizing and delivering of such resources. This also allows for greater job opportunities for newcomers, which as many participants indicated, can often be a challenge especially for young newcomers.

One of the main strengths of our study was that our participant recruitment and engagement strategy allowed us to engage youth leaders more deeply within our research team. These youth leaders played a critical role in recruiting participants from Africa and the Middle East, a challenge that previous research have documented [23]. We believe our openness to a range of stories was important to ensure participant autonomy. Based on the participants' feedback regarding experience with our study, there was some indication that giving them a voice and an opportunity to help others had a positive impact on participants. Sharing their stories may have also helped them better understand and adapt to a new world and left them in a better position to be supportive to other newcomers; i.e., peer support and future equity researchers.

The major limitation to our narrative method was the inability to enter in a dynamic ongoing two-way dialogue with participants to better understand their stories. For example, having the ability to fully contextualize experiences in the words of the participants may have added additional depth to the interpretations. Furthermore, our study did not include enough participants to allow for analysis of subgroups such as gender, country of origin, and immigration status. We also received more female than male participants, limiting our understanding of the male perspective. Finally, our participants needed to speak English or French, have access to the internet and have the ability to write a story, as well as already have connections with a local institution, which would have limited the breadth of perspectives. For example, most of our participants were students, and we recognized this link was an essential support for our participants. Broadening the scope of community leaders and affiliated local institutions for recruitment may help address these limitations in future studies. However, as ensuring a sense of community and security among participants is crucial in participatory research, incorporating in-person data collection methodologies may be more beneficial when working with larger groups of participants and to achieve a greater depth of understanding newcomers' stories. In this study, we have addressed the need for research based on social justice principles through the application of PHR. Our approach has allowed us to generate scientific knowledge aimed at advancing health equity and accessibility for marginalized populations. We found that winter weather conditions, language and cultural barriers, and perceived limited resources negatively impacts the integration and well-being of Canadian newcomers. In contrast, community building, social networks, and access to local resources facilitated the integration process. Further investigation of how immigration status, pre-settlement contexts, and gender can impact the integration journeys of newcomers in Canada could help to provide a more comprehensive understanding of their experiences. Future immigrant and refugee mental health research, as well as the development of culturally adaptive resources, should prioritize these key elements.

## Supporting information

**S1 Table.**
(PDF)

## Acknowledgments

We would like to acknowledge our participants for giving their time and sharing their stories.

## Author Contributions

**Conceptualization:** Kevin Pottie.

**Data curation:** Maggie Fong, Shahab Sayfi.

**Formal analysis:** Maggie Fong, Amy Liu, Bryan Lung, Ibrahim Alayche, Shahab Sayfi, Marie Hélène Chomienne, Ammar Saad, Jean Grenier, Azaad Kassam, Rukhsana Ahmed, Kevin Pottie.

**Funding acquisition:** Kevin Pottie.

**Investigation:** Maggie Fong, Amy Liu, Bryan Lung, Ibrahim Alayche.

**Methodology:** Kevin Pottie.

**Project administration:** Kevin Pottie.

**Resources:** Shahab Sayfi, Ryan Yuhi Kirenga, Azaad Kassam, Rukhsana Ahmed.

**Supervision:** Kevin Pottie.

**Visualization:** Maggie Fong, Amy Liu, Bryan Lung, Kevin Pottie.

**Writing – original draft:** Maggie Fong, Amy Liu, Bryan Lung, Ibrahim Alayche, Marie Hélène Chomienne, Jean Grenier, Kevin Pottie.

**Writing – review & editing:** Maggie Fong, Amy Liu, Bryan Lung, Ibrahim Alayche, Shahab Sayfi, Ryan Yuhi Kirenga, Marie Hélène Chomienne, Ammar Saad, Jean Grenier, Azaad Kassam, Rukhsana Ahmed, Kevin Pottie.

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
