## [Decision Letter · Decision Letter 0]

23 Oct 2023

PONE-D-23-29946From struggle to strength in African and Middle Eastern newcomers’ integration stories to Canada: A participatory health research studyPLOS ONE

Dear Dr. Pottie,

Thank you for submitting your manuscript to PLOS ONE. After careful consideration, we feel that it has merit but does not fully meet PLOS ONE’s publication criteria as it currently stands. Therefore, we invite you to submit a revised version of the manuscript that addresses the points raised during the review process.Your manuscript requires revision as mentioned by our reviewers. Please submit the revised manuscript.

We look forward to receiving your revised manuscript.

Kind regards,

AKM Alamgir, PhD

Academic Editor

PLOS ONE

Journal Requirements:

Reviewers' comments:

Reviewer's Responses to Questions

**Comments to the Author**

1. Is the manuscript technically sound, and do the data support the conclusions?

Reviewer #1: Yes

Reviewer #2: Yes

2. Has the statistical analysis been performed appropriately and rigorously? 

Reviewer #1: N/A

Reviewer #2: Yes

3. Have the authors made all data underlying the findings in their manuscript fully available?

Reviewer #1: No

Reviewer #2: Yes

4. Is the manuscript presented in an intelligible fashion and written in standard English?

Reviewer #1: Yes

Reviewer #2: Yes

5. Review Comments to the Author

Reviewer #1: Thank you for the opportunity to review the manuscript. The research is important and provides an important contribution to young adult immigrant and refugee experiences in Canada. What is less clear is the attention to mental health and health considerations and why they are not discussed in or referred back to in the discussion sections. It will be helpful to get some policy recommondations given the attention or the need for adaptive and culturally responsive health care access in the introduction. This needs to be followed through in the findings and discussion and then recommendations.

The participants were purposefully sampled but you note they were "intentionally found" I would recommend purposefully sampled as that is related to qualitative methods of recruitment.

Was it intentional to say that particpants were from Africa and the Middle East? these are continents and not countries for example; and this may reinforce a homogenous reading of the diversity of your participants. If this was done to protect your participants please explain.

Was informed consent obtained verbally as well as in writing and was it continous? you may want to see resources on rights of people in context of forced migration by "your rights in research by Canadian Council of Refugees"

In qualitative research member checking is not or should not be a form of validation only, this is because the process was reflexive related to the fact that you had youth/peers working with you and writing with you. How do you know people feel the same when you go back? it is process and so a few sentences about researcher reflexivity may help.

Similarly, data saturation is troubled as being 'a gold standard' for determining sample size of event the end of thematic analysis; consider concepts such as informational power.

What are policy recommendations moving forward that could included a gender based + analysis related to newcomer youth/resiliency? e.g. you mention local geography as being a contributor to loneliness and isolation but there were multiple factors. I appreciate the nod to a resiliency framework but what are the policy recommendations related to supporting the mental health and wellbeing of these newcomers?

Line 143-144 did you collect sociodemographic data related to Canadian Bench Marks for Language proficiency?

11% of your sample were refugees (people forced to migrate) do you know which streams does it matter for your overall findings and analysis? e.g. are the challenges uniquely different?

Finally, as the research is taking a participatory approach at what stages was consent received and how long did you engage with the participants and peer researchers? what were the impacts to hearing the stories and their mental health and wellbeing , e.g. potentially being triggered. Where there some limitations?

Line 352 you cite Canadian studies but only list one.

Overall the findings are important and relevant, greater attention to the above mentioned details and policy recommendations would be helpful. The most important being international students and how they might require support for their mental health and wellbeing in academic institutions.

Thank you for the opportunity to review

Reviewer #2: The study examined the experiences of African and Middle Eastern newcomers (immigrants, refugees, and international students) in Ontario, Canada, to better understand their integration challenges, successes, and social justice issues, aiming to enhance health and social services. Using a participatory research method, the authors collected stories from 18 newcomers (78% female, about half post-secondary students) aged 18-30 from Africa and the Middle East through snowball sampling within the immigrant community network. Qualitative narrative analysis was employed to interpret the stories, uncovering context themes, barriers, and facilitators. The authors found support from family, friends, education, and community services. The study highlights the significance of community building, friendships, and local services in immigrant service research. The paper addresses an intriguing subject of significant importance.

Overall, the paper was interesting and a nice addition to existing qualitative research on

newcomer integration. It’s an important topic and presenting this research using thematic content was a unique way to shed light on what newcomers find useful or lacking throughout the integration process. The discussion section was solid and well-written, however, the background and methods/results seemed a big vague in some areas. Please see below for specific comments:

Background (Lines 65-67): Why are newcomers particularly vulnerable? Could this be elaborated on? I also wondered what makes younger newcomers susceptible in comparison to older newcomers. This seems counterintuitive, as I expect it to be the reverse. Research has shown that younger newcomers are generally more flexible and adaptable. Is it context-specific?

Line 144: the percentage reported in brackets (88.9%) reads oddly and should perhaps be

moved slightly after the word “language.”

Methods/Results:

- What was the average range of ages of the participants?

- How long had the participants been in Ontario?

- It would be helpful if the authors gave more information about what was communicated with the participants regarding sharing a written story about their integration experience and journey. Were there any prompts that they were given? What were they instructed?

- Only two refugees were sampled for the study. Were their stories captured in the themes that originated from students and immigrants?

- The authors mentioned that they agreed on the overall themes but didn’t mention

how they went about/agreed on coding the stories. Also, they mention having four categorical themes, but it appears as though each category has subthemes, however, there is no mention of this in the methods section.

- Although immigrants, refugees, and international students are all newcomers, they are distinct groups with experiences and challenges. Whereas refugees are forced to leave their home countries due to circumstances beyond their control, immigrants choose to move for various reasons. On the other hand, international students come to study in a foreign country with the primary intention of pursuing education, often with the expectation of returning to their home country once their studies are completed. In Canada, international students are not considered immigrants, as their status is temporarily tied to their educational pursuits. It would be important to clearly distinguish between the three groups and to discuss if different themes emerged within each category.

- The authors may want to consider including a frequency table as it would be useful in providing a breakdown of which themes/subthemes populated often/less often between participant stories (unless all themes applied to every story, but this should be commented on either way for better clarification.)

Line 232 onwards: (iv) is coded as lack of resources and unknown scary migration context. The

former is not elaborated much in this section; it felt vague until I read the next section, making

me wonder if the sections should be combined.

Future Direction: The authors mention some important limitations, such as language and the

fact that all participants were students (though having a study on students’ experiences

is interesting.) Since the participants arrived from different countries, I did wonder about pre-

settlement contexts and differences between where an individual comes from and how this

might impact their settlement experience in Canada. I think this is beyond the scope of the

paper but would be maybe something they’d want to consider including under

conclusions/future direction.

6. PLOS authors have the option to publish the peer review history of their article (what does this mean?). If published, this will include your full peer review and any attached files.

Reviewer #1: No

Reviewer #2: No

---

## [Author Response · Author response to Decision Letter 0]

7 Jan 2024

Dear reviewers, 

Many thanks for the constructive feedback you provided. We integrated the comments in the new submitted manuscript. 

Thanks for your kind consideration, 

Corresponding author, 

Dr. Kevin Pottie 

Thank you for the opportunity to review the manuscript. The research is important and provides an important contribution to young adult immigrant and refugee experiences in Canada. What is less clear is the attention to mental health and health considerations and why they are not discussed in or referred back to in the discussion sections. It will be helpful to get some policy recommendations given the attention or the need for adaptive and culturally responsive health care access in the introduction. This needs to be followed through in the findings and discussion and then recommendations. 

Response: Thank you for your helpful review. We have revised to add more discussion on mental health implications and health considerations. Specifically, we discussed our results in relation to the provision of mental health resources for newcomers and policy recommendations regarding the structure of mental health and community organizations. 

The participants were purposefully sampled but you note they were "intentionally found" I would recommend purposefully sampled as that is related to qualitative methods of recruitment. 

Response: Thank you for the recommendation, edits have been made to the manuscript. 

Was it intentional to say that particpants were from Africa and the Middle East? these are continents and not countries for example; and this may reinforce a homogenous reading of the diversity of your participants. If this was done to protect your participants please explain. 

Response: Thanks. The use of “Africa and the Middle East” was intentional in the reporting of results. Information on the country of origin of participants was collected. Responses ranged throughout both continents; however, many participants originated from similar regions. Therefore, such information was removed from the manuscript and stories to protect the participants' identity as many conflict-affected regions have a small close-knit relationship and would risk being identified through some of the chosen quotes. Similarly, because snowball sampling was used the likelihood of participants knowing one another was higher, and thus also the risk of participant identification. Within our CBPR approach, we prioritized trust and safety for our participants to support safe story sharing. 

Was informed consent obtained verbally as well as in writing and was it continuous? you may want to see resources on rights of people in context of forced migration by "your rights in research by Canadian Council of Refugees" 

Response: Informed consent was obtained in writing at the start of the survey and story collection process. Participants were also aware that their consent could be withdrawn at any point throughout the study process. Participants who completed and submitted the online survey, which contained their written story, were provided with an honorarium for their time. We have edited our methods to make these points clear. 

In qualitative research member checking is not or should not be a form of validation only, this is because the process was reflexive related to the fact that you had youth/peers working with you and writing with you. How do you know people feel the same when you go back? it is process and so a few sentences about researcher reflexivity may help. 

Response: Our team consisted of a diverse group of authors who often played different roles throughout our study process. In that, youth who wrote for the project played less of a role in the analysis procedure until after all analyses had been completed. As well, participant names and identifying information were removed from stories prior to thematic analyses to ensure confidentiality, but also help minimize researcher reflexivity. We have added a section regarding details of our co-researchers and reflexivity in our methods section. 

Similarly, data saturation is troubled as being 'a gold standard' for determining sample size of event the end of thematic analysis; consider concepts such as informational power. 

Response: Thank you for this reference to information power. Authors coded and analyzed all stories individually. Themes that were prevalent in the majority of participants’ stories and that were coded by almost all authors for a story were represented in the results. A supplementary document including our study’s minimal underlying data has since been included for additional reference to mitigate harmful information power. 

What are policy recommendations moving forward that could included a gender based + analysis related to newcomer youth/resiliency? e.g. you mention local geography as being a contributor to loneliness and isolation but there were multiple factors. I appreciate the nod to a resiliency framework but what are the policy recommendations related to supporting the mental health and wellbeing of these newcomers? 

Response: This has been addressed in our previous response to comment #1. Specifically, we discuss the recommendation of providing young newcomers with leadership opportunities and roles in community-based support and organizations for other newcomers. This not only allows newcomers to feel more connected and less isolated, but also provides them with more job opportunities to ensure their basic needs are met. According to the resiliency framework, it is only once basic needs can be met, that newcomers are able to build and sustain resiliency. 

Line 143-144 did you collect sociodemographic data related to Canadian Bench Marks for Language proficiency? 

Response: Participants were asked to score their English and French proficiencies as “basic”, “intermediate”, “proficient”, “Not applicable (if it is their first language)” or they “preferred not to answer”. Most participants were more proficient in English compared to French. Participants were also asked to indicate the language they prefer communicating in when it comes to receiving or engaging in mental health services? The options included “English” , “French” , “Other” and “prefer not to answer”, and most participants responded “English” as well. 

11% of your sample were refugees (people forced to migrate) do you know which streams does it matter for your overall findings and analysis? e.g. are the challenges uniquely different? 

Response: Stories were deidentified prior to researchers analyzing and coding them. This was done to respect participant confidentiality as well as to minimize researchers’ confirmation bias when coding and interpreting them. Matching the results retrospectively, participants expressed similar challenges despite belonging to different categories of immigrant. The context of the challenges sometimes differed slightly, but not extensively. For example, those who are immigrant or refugee discussed loneliness and isolation more commonly in the context of work and community whereas international students discussed the challenge more in the context of academic environments. Our project was not large enough to conduct matrix analysis, for example in regard to refugee status, we have added 'refugees’ as a future research topic. 

Finally, as the research is taking a participatory approach at what stages was consent received and how long did you engage with the participants and peer researchers? what were the impacts to hearing the stories and their mental health and wellbeing , e.g. potentially being triggered. Where there some limitations? 

Response: Please refer to the response of comment #4 regarding consent. Participants and peer researchers were engaged with this project for a year; however, some played a more consistent role than others. As well, participants were able to discontinue the project at any point they felt uncomfortable or were no longer able to contribute. Only selective quotes and findings that were relevant to our results were shared with the majority of participants to avoid potential triggers and negative impacts on mental health. Like participants, researchers were also offered resources and support as needed. However, most researchers were familiar with this area of work and topic, so there were minimal disturbances. Instead, some participants and researchers actually expressed a feeling of empowerment after reading the stories. 

Line 352 you cite Canadian studies but only list one. 

Response: The Canadian study cited is a literature review that “aimed to identify risk factors and strategies in the approach to mental health assessment and to prevention and treatment of common mental health problems for immigrants in primary care”. Their results include and discuss the findings of 113 Canadian studies, 10 of which are systematic reviews and 3 that are meta-analyses. 

Background (Lines 65-67) :Why are newcomers particularly vulnerable? Could this be elaborated on? I also wondered what makes younger newcomers susceptible in comparison to older newcomers. This seems counterintuitive, as I expect it to be the reverse. Research has shown that younger newcomers are generally more flexible and adaptable. Is it context-specific? 

Response: A key factor contributing to the vulnerability (or precarity) of younger newcomers is the lack of tailored resources for youth. Many existing resources for immigrants and refugees are designed with families or middle-aged adults in mind, overlooking the unique needs and challenges faced by younger individuals. Furthermore, our results emphasize the pressing need for better resources, as evidenced by the numerous accounts from youth expressing their struggles and the necessity for more targeted support. 

Line 144: the percentage reported in brackets (88.9%) reads oddly and should perhaps be moved slightly after the word “language.” 

Response: Thank you for the recommendation, revisions have been made to the manuscript. 

What was the average range of ages of the participants? 

Response: The range of ages of the participants was 18-30. 

How long had the participants been in Ontario? 

Response: “Newcomers were defined as those who have been here between 1-10 years.” 

It would be helpful if the authors gave more information about what was communicated with the participants regarding sharing a written story about their integration experience and journey. Were there any prompts that they were given? What were they instructed? 

Response: Participants were provided with an e-learning module on storytelling, which outlined prompts and story examples to guide participants. We have included this e-learning module in our supporting information. 

Only two refugees were sampled for the study. Were their stories captured in the themes that originated from students and immigrants? 

Response: See our response of comment #9- also on refugees> 

“Response: Stories were deidentified prior to researchers analyzing and coding them. This was done to respect participant confidentiality as well as to minimize researchers’ confirmation bias when coding and interpreting them. Matching the results retrospectively, participants expressed similar challenges despite belonging to different categories of immigrant. The context of the challenges sometimes differed slightly, but not extensively. For example, those who are immigrant or refugee discussed loneliness and isolation more commonly in the context of work and community whereas international students discussed the challenge more in the context of academic environments. Our project was not large enough to conduct matrix analysis, for example in regard to refugee status, we have added 'refugees’ as a future research topic.” 

The authors mentioned that they agreed on the overall themes but didn’t mention how they went about/agreed on coding the stories. Also, they mention having four categorical themes, but it appears as though each category has subthemes, however, there is no mention of this in the methods section. 

Response: Thank you. We have clarified this in our methods section. Each author independently read and annotated the stories. Common themes were then discussed and collaboratively agreed upon, and these were subsequently categorized under the four overarching themes. Disagreements in interpretation were resolved through extensive discussions within the larger research team. Revisions have been made to the manuscript to provide a more comprehensive understanding of our approach. 

Although immigrants, refugees, and international students are all newcomers, they are distinct groups with experiences and challenges. Whereas refugees are forced to leave their home countries due to circumstances beyond their control, immigrants choose to move for various reasons. On the other hand, international students come to study in a foreign country with the primary intention of pursuing education, often with the expectation of returning to their home country once their studies are completed. In Canada, international students are not considered immigrants, as their status is temporarily tied to their educational pursuits. It would be important to clearly distinguish between the three groups and to discuss if different themes emerged within each category. 

Response: Thank you. While we understand there are different groups of newcomers, our intention was to focus on the age-related lived experiences of youth. Incidentally in Canada, it is well recognized that most international students will have a direct line for immigration. It is also possible that an international student may be fleeing conflict in a different way. Participants were not categorized based on these distinctions, as many participants in our study belonged to multiple groups (i.e. despite being a refugee, they spent the majority of their life in Canada as a student; therefore, also experiencing the struggles and facilitators of the Canadian education system). Authors were also blinded to participants' details to prevent bias, recognizing that preconceived expectations could influence the focus on particular challenges in the stories of each newcomer population. 

The authors may want to consider including a frequency table as it would be useful in providing a breakdown of which themes/subthemes populated often/less often between participant stories (unless all themes applied to every story, but this should be commented on either way for better clarification.) 

Response: Thank you for the recommendation. We have included a table of the populated themes and quotes that was coded for in our supporting documents. 

Line 232 onwards: (iv) is coded as lack of resources and unknown scary migration context. The former is not elaborated much in this section; it felt vague until I read the next section, making me wonder if the sections should be combined. 

Response: Integrated in the manuscript. 

Future Direction: The authors mention some important limitations, such as language and the fact that all participants were students (though having a study on students’ experiences is interesting.) Since the participants arrived from different countries, I did wonder about pre- settlement contexts and differences between where an individual comes from and how this might impact their settlement experience in Canada. I think this is beyond the scope of the paper but would be maybe something they’d want to consider including under conclusions/future direction. 

Response: We have integrated this point in the manuscript. Overall this is out of the scope our research, but we agree it is important for future research.

---

## [Decision Letter · Decision Letter 1]

6 Feb 2024

PONE-D-23-29946R1From struggle to strength in African and Middle Eastern newcomers’ integration stories to Canada: A participatory health equity research studyPLOS ONE

Dear Dr. Pottie,

Thank you for submitting your manuscript to PLOS ONE. After careful consideration, we feel that it has merit but does not fully meet PLOS ONE’s publication criteria as it currently stands. Therefore, we invite you to submit a revised version of the manuscript that addresses the points raised during the review process by Mar 22 2024 11:59PM..

We look forward to receiving your revised manuscript.

Kind regards,

AKM Alamgir, PhD

Academic Editor

PLOS ONE

Reviewers' comments:

Reviewer's Responses to Questions

**Comments to the Author**

1. If the authors have adequately addressed your comments raised in a previous round of review and you feel that this manuscript is now acceptable for publication, you may indicate that here to bypass the “Comments to the Author” section, enter your conflict of interest statement in the “Confidential to Editor” section, and submit your "Accept" recommendation.

Reviewer #2: All comments have been addressed

Reviewer #3: (No Response)

2. Is the manuscript technically sound, and do the data support the conclusions?

Reviewer #2: Yes

Reviewer #3: Yes

3. Has the statistical analysis been performed appropriately and rigorously? 

Reviewer #2: N/A

Reviewer #3: N/A

4. Have the authors made all data underlying the findings in their manuscript fully available?

Reviewer #2: Yes

Reviewer #3: Yes

5. Is the manuscript presented in an intelligible fashion and written in standard English?

Reviewer #2: Yes

Reviewer #3: Yes

6. Review Comments to the Author

Reviewer #2: (No Response)

Reviewer #3: It is with interest that I have read this manuscript in its revised version. This research adds important understanding to the challenges that face newcomers from Africa and the Middle East and points out opportunities to facilitate their integration and support building their resilience. The results set a foundation for future research and initiatives to improve health equity and social services tailored to this marginalized population.

The authors utilized appropriate relevant literature to support their approach, including those that justify their predetermined age range of recruits between 18-30. Moreover, the need to run such research and its implication for improving the welfare of such communities was properly described, which made the objective clear. The selected type of research was also properly justified based on previous literature.

The authors clearly described their methodology as well as their strategic involvement of newcomer researchers to help build the “trust” needed for the input of the participants. However, in response to the first reviewer inquiry regarding any prompts or instructions given to the participants, the authors explained that “Participants were provided with an e-learning module on storytelling, which outlined prompts and story examples to guide participants. We have included this e-learning module in our supporting information.” I would suggest mentioning this in the methodology as it gives valuable information on how the stories were collected from the participants, which would guide future research.

In line [115], the statement is an exact repetition of the opening statement in line [86]. If the authors find the need to mention the approach again in that section, I suggest rephrasing it to avoid redundance.

In the Results section, the added table was useful. I would suggest replacing “Count” by “Frequency (n=18)” and replace “Frequency” with “Percentage”. Also, in the table section (Highest Level of Education Completed), the total is 19 which suggests an error to be corrected.

In lines [342-344] of the Discussion section, I found the statement a bit unclear. I had to read it several times to understand that the authors found that the amount of literacy, trust, resilience, and self-confidence were attributes that could enable emerging leaders. I suggest a simple rephrasing to avoid the unclarity of an important statement.

In lines [426-435], the authors discuss the limitations of their research, including the recruitment process. It might be useful if they can suggest recommendations to overcome this limitation in future research based on their experience, which would be beneficial for future researchers building on their methodology.

7. PLOS authors have the option to publish the peer review history of their article (what does this mean?). If published, this will include your full peer review and any attached files.

Reviewer #2: No

Reviewer #3: No

---

## [Author Response · Author response to Decision Letter 1]

24 Mar 2024

It is with interest that I have read this manuscript in its revised version. This research adds important understanding to the challenges that face newcomers from Africa and the Middle East and points out opportunities to facilitate their integration and support building their resilience. The results set a foundation for future research and initiatives to improve health equity and social services tailored to this marginalized population.

The authors utilized appropriate relevant literature to support their approach, including those that justify their predetermined age range of recruits between 18-30. Moreover, the need to run such research and its implication for improving the welfare of such communities was properly described, which made the objective clear. The selected type of research was also properly justified based on previous literature.

The authors clearly described their methodology as well as their strategic involvement of newcomer researchers to help build the “trust” needed for the input of the participants. However, in response to the first reviewer inquiry regarding any prompts or instructions given to the participants, the authors explained that “Participants were provided with an e-learning module on storytelling, which outlined prompts and story examples to guide participants. We have included this e-learning module in our supporting information” I would suggest mentioning this in the methodology as it gives valuable information on how the stories were collected from the participants, which would guide future research.

Response: Thank you for your helpful review. We have incorporated your suggestion in the methodology of our revised manuscript. 

In line [115], the statement is an exact repetition of the opening statement in line [86]. If the authors find the need to mention the approach again in that section, I suggest rephrasing it to avoid redundance.

Response: Integrated in the manuscript.

In the Results section, the added table was useful. I would suggest replacing “Count” by “Frequency (n=18)” and replace “Frequency” with “Percentage”. Also, in the table section (Highest Level of Education Completed), the total is 19 which suggests an error to be corrected.

Response: Thank you for your recommendation and for bringing this error to our attention. Revisions have been made to the manuscript.

In lines [342-344] of the Discussion section, I found the statement a bit unclear. I had to read it several times to understand that the authors found that the amount of literacy, trust, resilience, and self-confidence were attributes that could enable emerging leaders. I suggest a simple rephrasing to avoid the unclarity of an important statement.

Response: Integrated in the manuscript. 

In lines [426-435], the authors discuss the limitations of their research, including the recruitment process. It might be useful if they can suggest recommendations to overcome this limitation in future research based on their experience, which would be beneficial for future researchers building on their methodology.

Response: Thank you for your recommendation. We have integrated this point in the manuscript.

---

## [Decision Letter · Decision Letter 2]

9 Apr 2024

From struggle to strength in African and Middle Eastern newcomers’ integration stories to Canada: A participatory health equity research study

PONE-D-23-29946R2

Dear Dr. Kevin Pottie,

We’re pleased to inform you that your manuscript has been judged scientifically suitable for publication and will be formally accepted for publication once it meets all outstanding technical requirements.

Kind regards,

AKM Alamgir, PhD

Academic Editor

PLOS ONE

Additional Editor Comments (optional):

Reviewers' comments:

Reviewer's Responses to Questions

**Comments to the Author**

1. If the authors have adequately addressed your comments raised in a previous round of review and you feel that this manuscript is now acceptable for publication, you may indicate that here to bypass the “Comments to the Author” section, enter your conflict of interest statement in the “Confidential to Editor” section, and submit your "Accept" recommendation.

Reviewer #3: All comments have been addressed

2. Is the manuscript technically sound, and do the data support the conclusions?

Reviewer #3: Yes

3. Has the statistical analysis been performed appropriately and rigorously? 

Reviewer #3: Yes

4. Have the authors made all data underlying the findings in their manuscript fully available?

Reviewer #3: Yes

5. Is the manuscript presented in an intelligible fashion and written in standard English?

Reviewer #3: Yes

6. Review Comments to the Author

Reviewer #3: It was interesting to see the final version of the manuscript with the final amendments. The authors ensured a smooth integration of the modifications and addressed the recommended suggestions.

We look forward to have future research built on this one to advance newcomer's health equity.

7. PLOS authors have the option to publish the peer review history of their article (what does this mean?). If published, this will include your full peer review and any attached files.

Reviewer #3: **Yes: **Madona Yahia
